# FROM CROWDS TO CODES: MINIMIZING REVIEW BURDEN IN CONFERENCE REVIEW PROTOCOLS

## ABSTRACT

Conference peer review aims to accurately assess paper quality while minimizing review load. This paper explores optimal conference protocols — rules for designing review tasks to reviewers and inferring paper quality based on the noisy review. The widely used *direct review* protocol assigns multiple independent reviewers to each paper in an *isolated* and *parallel* manner. However, as submission volumes grow, more complex protocols have developed, e.g., two-phase review and meta-review.

In this paper, we investigate whether and when these more complex *joint* and *adaptive* protocols can reduce the *review load ratio*, the number of review tasks per paper. Using tools from information theory and coding theory, we establish the following results:

- We prove that the optimal load ratio for isolated protocols is $\Theta(\ln n/\epsilon)$, where $n$ is the number of papers and $\epsilon$ is the error probability indicating that the review load ratio increases as the number of papers grows.
- We prove that the optimal load ratio of joint protocols is a constant dependent on the agents' noise levels and independent of both $n$ and $\epsilon$. This suggests that joint protocols—including two-phase review—can dramatically reduce the review burden.
- We empirically explore the design of two-phase review protocols and find that selecting the borderline (ambiguous) papers for the second phase review can significantly increase the accuracy compared to the conventional selection of a better fraction of promising papers for the second phase.

## 1 INTRODUCTION

Peer review, the process of evaluating scientific research by volunteer experts, is critical in ensuring the quality of accepted papers. However, the rapid growth of submissions has placed growing pressure on the peer review system. Sculley et al. (2018); Shah (2019) Conferences face two often competing objectives: minimizing estimation error in inferring paper quality while reducing review load. Despite a growing body of research on conference design, relatively little work has focused on designing review tasks to optimize these two objectives. This work initiates the study of conference review protocols—rules for designing review tasks for reviewers (review protocol) and inferring paper quality based on the noisy review (inference protocol)—with the goal of minimizing review load while ensuring a given accuracy guarantee.

Traditional conferences assign multiple agents to directly review each paper to ensure an accurate assessment of paper quality. Recently, many alternative protocols have emerged. These include meta-review tasks, where senior reviewers aggregate multiple reviews into a single review; two-phase protocols Leyton-Brown et al. (2022), where second-phase reviews depend on the first phase; and comparison-based tasks, where reviewers compare multiple papers. Or even protocols that consist of multiple different review tasks. These examples raise a fundamental question: What is the optimal conference review protocol?

To answer these questions, we first classify conference review protocols based on two key attributes—isolation and adaptivity—illustrated in Fig. 1. First, isolation captures whether a review task depends solely on a single paper. Examples of isolated protocols include direct review

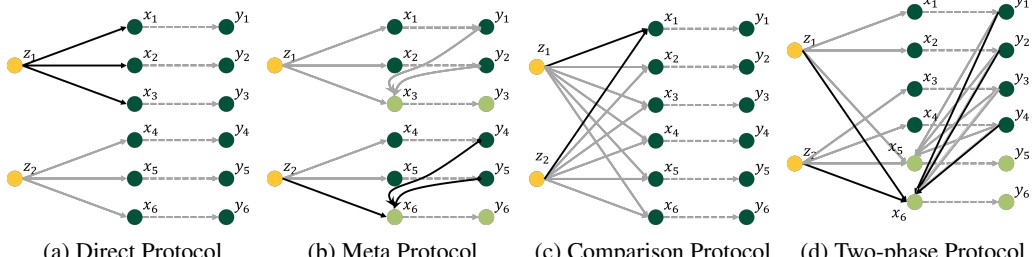

Figure 1: Examples of review protocols and tasks: In the direct protocol Fig. 1a (isolated and nonadaptive), each paper receives three direct reviews. In the meta-review protocol Fig. 1b (isolated and adaptive), each paper receives two direct reviews plus one meta-review (light green) that depends on the direct reviews. In the comparison protocol Fig. 1c (joint and nonadaptive), each task depends on multiple papers. In the two-phase protocol Fig. 1d (joint and nonadaptive), each paper receives two direct reviews in Phase 1, and the Phase 2 review (light green) depends on all first-phase reviews and on multiple papers.

or naive meta-reviews which only depends on reviews of a single paper. On the other hand, joint protocols (non-isolated) contain two-phase reviews or comparison-based reviews. In particular, in a two-phase protocol, if only the top 50% of the first phase paper can enter the second phase, the review tasks in the second phase inherently depend on the rankings from the first phase. Kozyrakis & Berger (2021) One of our main results shows that joint review protocols can significantly outperform isolated review protocols to reduce error probability under the same review load.

The second attribute—adaptivity—refers to whether a review task depends on the outcomes of other review tasks. Examples of parallel (non-adaptive) protocols include direct review and comparison-based tasks, while adaptive protocols, such as meta-reviews and two-phase protocols, assign reviews based on previous evaluations. We show that that adaptive review protocols do not necessarily reduce error probability under the same review load compared to parallel ones in both isolated or joint settings.

**Technical contributions**    Given $n$ papers, $\epsilon > 0$, and known noise levels $\mathcal{D}_q$, we study the optimal conference review protocol that minimizes the *review load ratio*, the number of review tasks per paper, while ensuring that the final estimation's *error probability* is at most $\epsilon$. Leveraging concepts from information theory and coding theory, we establish a strong connection between coding theory and the conference review problem. Through this connection, we derive the following key results:

- We prove that the optimal load ratio for isolated protocols is $\Theta(\ln n/\epsilon)$ in Theorem 4.1, where $n$ is the number of papers and $\epsilon$ is the error probability. To prove Theorem 4.1, we show that direct review protocols Blackwell dominate any other isolated adaptive protocol in Lemma 4.2.

- We prove that the optimal load ratio of joint protocols is a constant $\lambda^*$ in Eq. (2) dependent on the agents' noise levels and independent of both $n$ and $\epsilon$. Interestingly, Theorem 4.5 can be viewed as a non-asymptotic version of the celebrated Shannon's noisy channel theorem. Additionally, Theorem 4.5 shows that parallel protocols are sufficient to minimize the review load ratio.

- Two-phase Review Protocols: In Section 5, we illustrate with empirical results that adaptive protocols can potentially reduce the load ratio in joint review processes. We show this by comparing simulation results of two-phase adaptive reviews and single-stage baselines.

**Conceptual contributions**    One notable aspect of our model is the explicit separation between review protocols and inference protocols. This distinction highlights an agent's contribution can be informational and computational. For example, most Program Committee (PC) members primarily provide informational contributions by directly reviewing papers. Senior Program Committee (SPC) members, Area Chairs (ACs), and Program Chairs also contribute computationally by designing review tasks (paper assignments, deciding PC and SPC) and summarizing reviews rather

than accessing new information. This distinction provides insight into how responsibilities could be allocated more effectively.

Another key result from our work is that joint protocols can outperform isolated protocols, underscoring the value of leveraging information across multiple papers. For example, as demonstrated in the empirical section, two-phase review protocols use early-stage information across submissions to improve the load ratio. This finding also suggests that, rather than merely summarizing individual reviews, SPCs or ACs could play a more impactful role by synthesizing and communicating a broader perspective across submissions—facilitating more effective comparisons between papers.

## 2 RELATED WORKS

Our work is primarily relevant to the papers that try to understand the conference design problem.

Several conference review protocols have been introduced to improve efficiency and decision accuracy. Prominent examples include two-phase review Kozyrakis & Berger (2021); Leyton-Brown et al. (2022) or summary or desk rejects Yuan (2020). Xie & Lui (2012) present a mathematical model to analyze various factors that may influence the accuracy of peer review systems. Their model predicts that three reviews per paper are sufficient for a high-accuracy decision, while this number increases to more than seven for prestigious conferences. Furthermore, a two-phase mixed design of reviewer assignment is proposed to reduce the review workload and improve the decision accuracy. In addition, there is a stream of literature studying using author's comparison among their papers to improve conference design Su (2021); Wu et al. (2023); Zhang et al. (2024). Our review protocol provide a mathematical formulation for the above protocols.

Our study is also relevant to studies on reviewer assignment problem Taylor (2008). Our conference protocol can be seen as generalization, where reviewer assignments can be seen as direct review protocols. The reviewer assignment problem considers the noise level when evaluating different papers, and focuses on developing novel assignment algorithms to better match qualified reviewers to papers. However, we assume the noise level is independent of assigned review tasks. This line of research usually considers maximizing an objective, such as the welfare, subject to fairness constraints Stelmakh et al. (2019); Payan & Zick (2021); Aziz et al. (2024), or mitigating manipulations Jecmen et al. (2020); Stelmakh et al. (2021a); Cohen et al. (2016). Furthermore, a substantial amount of theoretical and empirical studies aim to improve peer the quality of peer review by understanding and mitigating review bias Lee et al. (2013); Haffar et al. (2019); Tomkins et al. (2017); Stelmakh et al. (2021b).

Different from the above discussions, our paper aims to develop theoretical insights with a focus on comparing the design of the review structure, i.e. how to distribute the review workload to various types of review tasks.

Ductor et al. (2020) present a model of journals as platforms to understand the publication trends across time and disciplines. Under their model, the authors observe that whether the field-specific journals or the general journal can publish the best papers depends on the number of fields and readership of the general journal. For example, one of their results suggests that the general journal attracts the best papers in equilibrium when the product of the number of fields times the readership of the general journal is large. Zhang et al. (2022) consider the interactions between a prestigious conference and a large group of strategic authors as a Stackelberg game, where authors decide whether to submit their papers to the conference. They show that the conference can achieve a better trade-off between its quality and the review burden when the review quality improves, the conference becomes less attractive, and the authors become more patient.

From the perspective of using coding theory to understand the task assignment on multi-agent platforms, the most relevant work is given by Vempaty et al. (2014). Their work uses error-control codes and decoding algorithms to design crowdsourcing systems for reliable classification given noisy crowds. Their proposed coding-base crowsouring schemes are shown to outperform the classic majority vote method. We emphasize that our work sharply deviates from the above work as we consider much more general applications of review protocols. In particular, our main focus is on comparing the direct review protocol with the more complex adaptive review protocol, while Vempaty et al. (2014) only consider several special types of crowdsourcing models.

## 3 CONFERENCE REVIEW PROBLEM

We study how to infer the quality of $n$ papers from $m$ review tasks. We introduce our model for conference problems and protocols in Section 3, and discuss extension in Section 6. Given a collection of papers and the noise levels in the agents' reports, a conference protocol consists of a review protocol and an inference protocol (defined in Section 3.1) to estimate paper quality. The performance is measured by load ratio and error probability.

We will use normal text to denote random objects and boldface to represent vectors. For example, x and **x** denote a random variable and a random vector, respectively, while $x$ and $\boldsymbol{x}$ represent their corresponding outcomes. Let $[n] := \{1, \ldots, n\}$.

A ***conference review problem*** is defined by the parameters $(n, \mathcal{D}_{\boldsymbol{q}})$ where $n$ is the number of papers and $\mathcal{D}_{\boldsymbol{q}}$ is the distribution over agents' noise levels. Each paper $j \in [n]$ has a binary true state $z_j \in \{-1, 1\}$ with uniform prior. The goal is to infer these true qualities from the agents' reports.

Agents are assigned *review tasks*, which serve as the basic unit of work in our model. For example, if 100 papers each requires 10 reviews, this results in $m = 1000$ tasks. We call $m$ as *review load* and $\lambda := \frac{m}{n}$ as *review load ratio*, quantifying the average review resources per paper. Each review task $i \in [m]$ consists of a binary question $f_i$ with a true answer $x_i \in \{-1, 1\}$ and an assigned agent submits a report $y_i \in \{-1, 1\}$ submitted by an agent. The reliability of the submitted report on task $i$ depends on the agent's *noise level* $q_i \in [0, 1/2]$ which models the probability of the agent's report disagreeing with the true answer. In particular, for all $x_i \in \{-1, 1\}$,

$$\text{y}_i = \begin{cases} x_i & \text{with probability } 1 - q_i \\ -x_i & \text{otherwise,} \end{cases} \tag{1}$$

where the noise is mutually independent for all $i \in [m]$. We denote these processes as $x_i \xrightarrow{q_i} \text{y}_i$. We use $\boldsymbol{q} = (q_1, \ldots, q_m)$ to denote the noise levels of all tasks and use the histogram $\mathcal{D}_{\boldsymbol{q}}$ to denote the distribution such that $m \, \mathcal{D}_{\boldsymbol{q}}(q)$ is the number of tasks with a noise level of $q$. Here we assume that the minimum and maximum of noise levels exist and separated from boundary, $0 < \min_{q \in supp(\mathcal{D}_q)}$ and $\max_{q \in supp(\mathcal{D}_q)} < 1/2$. Note that the ordering of $\boldsymbol{q}$ is irrelevant to the design of the optimal protocol, as tasks can be permuted. The noise level can also be interpreted as the probability of shirking, and agents who put in effort on a given task are correspondingly more likely to report the true answer.

Given a conference review problem $(n, \mathcal{D}_{\boldsymbol{q}})$, a ***conference protocol*** with load ratio $\lambda$ operates through two steps: a review protocol $\boldsymbol{f}$ and an inference protocol $g$:

- Review protocol: Design $m = \lambda n$ review tasks $\boldsymbol{f} = (f_1, \ldots, f_m)$, and agents submit reports $\boldsymbol{y} = (y_1, \ldots, y_m) \in \{-1, 1\}^m$ based on the noise levels from the histogram $\mathcal{D}_{\boldsymbol{q}}$.
- Inference protocol: Infer the quality of $n$ papers, $g(\boldsymbol{y}) = \hat{\boldsymbol{z}} = (\hat{z}_1, \ldots, \hat{z}_n) \in \{-1, 1\}^n$, using the collected reports $\boldsymbol{y}$.

The conference cares about the *error probability*, denoted as $P_e = \Pr[\hat{\mathbf{z}} \neq \mathbf{z}]$, where the randomness arises from $\mathbf{z}$, which is uniformly distributed, and the reviewer's noise. Our goal is to design a conference protocol $(\boldsymbol{f}, g)$ that minimizes the load ratio $\lambda$ while ensuring an almost zero error probability. Given $\mathcal{D}_{\boldsymbol{q}}$ and $n$, a conference protocol $(\boldsymbol{f}, g)$ is a $(\lambda, \epsilon)$-protocol if the load ratio is upper bounded by $\lambda$ and error probability is upper bounded by $\epsilon$. We further empirically investigate different metric, e.g., average accuracy and calibration in Section 5.

### 3.1 REVIEW AND INFERENCE PROTOCOLS

A review protocol specifies the design of review tasks. We classify the review protocols based on two key attributes: isolation and adaptiveness. A protocol is isolated if each review task pertains to a single paper. Most common review protocols are isolated. This includes the *direct review protocol*, where each reviewer evaluates a paper independently, and the *meta-review protocol*, where a meta-reviewer assesses multiple reviewers' judgments of the same paper. A protocol is non-adaptive (or parallel) if review tasks are independent of each other and the reports are simultaneously collected. For example, the direct review protocol is parallel while the meta-review protocol is isolated but adaptive. Fig. 1 illustrate examples of each class of review protocols.

**Isolated review protocols**   One of the simplest tasks is to directly evaluate whether a paper is of good quality. Specifically, if task $i$ is a *direct review task* of paper $j$, the report is a noisy estimate of paper $j$'s quality

$$y_i \xleftarrow{q_i} f_i(\boldsymbol{z}) = z_j$$

and the true answer is $x_i = z_j$. A review protocol that consists entirely of direct review tasks is called the *direct review protocol*. Each direct review protocol can be represented as a function $\sigma : [m] \to [n]$ where task $i$ is a direct review of paper $\sigma(i)$, with $x_i = z_{\sigma(i)}$. This setup aligns with the common reviewer assignment problem Jovanovic & Bagheri (2023), where each reviewer evaluates a subset of papers and reports their evaluations for all papers in the subset.

Another example is to design a task that checks whether a previous report is correct. Consider the review of a single paper $n = 1$ whose true quality is $z$. The first task is a direct review task $f_1(z) = z$ and the $(i + 1)$-th task checks the correctness of the $i$-th report $y_i$ for $i > 1$,

$$y_{i+1} \xleftarrow{q_{i+1}} f_{i+1}(z, y_1, \dots, y_i) = \begin{cases} 1 & \text{if } f_i(z, y_1, \dots, y_{i-1}) \neq y_i, \\ -1 & \text{if } f_i(z, y_1, \dots, y_{i-1}) = y_i. \end{cases}$$

We refer to such tasks as *successive review tasks* Abramowitz et al. (2023), and a review protocol is called a successive review protocol if it consists entirely of chains of successive review tasks for each paper. Notably, a successive review protocol is *isolated*, as each task pertains to a single paper, and *adaptive*, as the design of each task may depend on the outcomes of previous tasks in the chain. Most conference designs adopt a *meta-review structure*, where a paper typically receives several direct reviews followed by a meta-review that summarizes these reviews. Specifically, a paper gets $i$ direct reviews and the $i + 1$-th task aggregates these direct reviews into a unified evaluation:

$$f_1(z) = \dots = f_i(z) = z, \text{ and } f_{i+1}(z, y_1, \dots, y_i).$$

More generally, for a single paper, the review protocol can form an arbitrary chain of $m'$ tasks $x_1 = f_1(z) \xrightarrow{q_1} y_1, x_2 = f_2(z, y_1) \xrightarrow{q_2} y_2, ..., \text{ and } x_{m'} = f_{m'}(z, y_1, \dots, y_{m'-1}) \xrightarrow{q_{m'}} y_{m'}$. We call a review protocol *isolated* if the review tasks can be partitioned into independent chains, with each chain corresponding to a single paper. All the above examples are isolated review protocols.

**Joint Review Protocol**   A review protocol and its tasks do not need to be isolated. For example, a review task may require reviewers to compare multiple papers. More generally, an adaptive and joint review protocol $(f_i)_{i \in [m]}$ consists of a sequence of tasks ordered by input dependencies. The reports $\boldsymbol{y}$ are generated through the following process:

$$\boldsymbol{z} \sim_u \{-1, 1\}^n, x_1 = f_1(\boldsymbol{z}) \xrightarrow{q_1} y_1, \dots, x_m = f_m(\boldsymbol{z}, y_1, \dots, y_{m-1}) \xrightarrow{q_m} y_m$$

Similarly, a review protocol can be parallel and joint if each review task is independent of the outcomes of other tasks.

Compared with the isolated protocol, the main difference here is that the review tasks can depend on the ground state of all papers $\boldsymbol{z}$ instead of a single paper. Examples of parallel tasks include deciding whether one paper is more likely to be accepted than another (Shah et al., 2017), determining if the number of high-quality papers within a subset is even, or whether all papers in a subset are good. The second is called an *XOR review task* on a subset of papers $S \subseteq [n]$, where the answer is given by $x = \oplus_{j \in S} z_j$, with $\oplus$ denotes the XOR operator. The third is called an *AND review task* Mazumdar & Pal (2017); Pang et al. (2019). These examples are parallel but joint.

Finally, the *two-phase review protocol* is adaptive and joint. The idea is to assign additional reviews to a paper if it is flagged for further evaluation after receiving borderline or conflicting reviews in the first phase. This protocol can be both adaptive and joint because the decision to assign additional reviews depends on the first-phase reviews of multiple papers.[1]

A conference protocol $(\boldsymbol{f}, g)$ is *deterministic* if $\boldsymbol{f}$ and $g$ are deterministic. We note that the optimal inference protocol can be deterministic. Given a review protocol $\boldsymbol{f}$, an inference protocol that minimizes the error probability $P_e$ is the maximum a posteriori estimator, or equivalently the maximal likelihood estimator (MLE) as the prior of paper is uniform.

---

[1]A two-phase protocol can be joint if only the top 50% of the first phase paper can enter the second phase as in Kozyrakis & Berger (2021), but can also be isolated if a paper can enter the second phase only depends on its own reviews in the first phase as in Leyton-Brown et al. (2022)

The following remarks a natural correspondence between designing codes on a channel and conference protocol on a conference problem. We provide a brief introduction to coding theory in the appendix.

**Remark 3.1** (Conference design as a coding problem). *Consider* w *as the underlying quality of* $n$ *papers, corresponding to the original message in coding theory. The conference cannot access the underlying quality directly, but can obtain information on it via* $m$ *review tasks. Each review task* $i$ *has a true answer* $x_i$ *and the reviewer's report on that task is* $y_i$, *which is a potentially distorted version of* $x_i$. *The reviewer noise can thus be interpreted as the channel noise. Finally, the conference aims to recover* w *as accurately as possible using the inference protocol, which maps to the decoder in coding theory.*

# 4  OPTIMIZING THE LOAD RATIO

We first show a tight bound on the optimal load ratio for isolated protocols. Specifically, the optimal load ratio of isolated protocols for small error probability $P_e = \epsilon$ is $\Theta(\ln \frac{n}{\epsilon})$.

**Theorem 4.1.** *Given a conference problem* $(n, \mathcal{D}_{\boldsymbol{q}})$ *and small enough* $\epsilon > 0$, *the optimal isolated protocol* $(\boldsymbol{f}, g)$ *with error probability* $P_e = \epsilon$ *requires load ratio* $\lambda = \Theta\left(\ln \frac{n}{\epsilon}\right)$.

The proof consists of two parts. We first show that the direct protocol is the optimal isolated protocol in the Blackwell sense in Lemma 4.2 and provide a tight bound on the optimal load ratio of direct review protocols in Lemma 4.4.

We say a protocol $(\boldsymbol{f}, g)$ *Blackwell dominates* $(\boldsymbol{f}', g')$ if the reports $\mathbf{y}$ generated by the former Blackwell dominate those of the latter $\mathbf{y}'$. The formal definition of Blackwell dominance is in the appendix. Intuitively, the report $\mathbf{y}$ from $\boldsymbol{f}$ can simulate report $\mathbf{y}'$ from $\boldsymbol{f}'$ via a garbling. Hence, if one protocol Blackwell dominates another, its error probability is always less than or equal to that of the other.

**Lemma 4.2.** *Given any conference review problem* $(n, \mathcal{D}_{\boldsymbol{q}})$, *for any isolated review protocol* $\boldsymbol{f}$, *there exists a direct review protocol* $\boldsymbol{f}'$ *with the same review load ratio that Blackwell dominates the isolated review protocol.*

Lemma 4.2 suggests that direct protocols are sufficient when the noise levels are task-independent. However, a more complex review protocol may suffer from higher noise levels. For instance, a meta-review task summarizing multiple conflicting reviews may have a higher noise level than a direct review.

To account for this factor, we further provide a complementary result showing that the Blackwell dominance relationship between two protocols is monotone with respect to review noise. In other words, if a review protocol $\boldsymbol{f}$ that is Blackwell dominated by another protocol $\boldsymbol{f}'$ under the same noise levels, then $\boldsymbol{f}$ remains dominated by $\boldsymbol{f}'$ as long as the noise in $\boldsymbol{f}$ decreases across all tasks. With Proposition 4.3, we establish a stronger result: if noise levels is non-decreasing in an adaptive protocol, our conclusion in Lemma 4.2 still holds.

**Proposition 4.3.** *Given* $m$ *and two conference problems* $(n, \mathcal{D}_{\boldsymbol{q}})$ *and* $(n, \mathcal{D}_{\boldsymbol{q}}')$ *if* $q_i \leq q_i'$ *for all* $i \in [m]$, *then for any review protocol* $\boldsymbol{f}$ *with load* $m$ *there exists a review protocol* $\boldsymbol{f}'$ *with the same review load so that the reports* $\mathbf{y}$ *from protocol* $\boldsymbol{f}$ *under* $(n, \mathcal{D}_{\boldsymbol{q}})$ *Blackwell dominates the reports* $\mathbf{y}'$ *from protocol* $\boldsymbol{f}'$ *under* $(n, \mathcal{D}_{\boldsymbol{q}}')$.

**Lemma 4.4.** *Given a conference problem* $(n, \mathcal{D}_{\boldsymbol{q}})$ *where* $\mathcal{D}_{\boldsymbol{q}} = q\mathbf{1}$ *for some constant* $0 < q < 1/2$ *and* $\epsilon > 0$, *the optimal isolated protocol* $(\boldsymbol{f}, g)$ *with error probability* $P_e = \epsilon$ *has load ratio* $\lambda = \frac{1}{\ln(1/4q(1-q))} \ln \frac{n}{\epsilon} + O(1)$ *as* $\epsilon \to 0$.

Given the characterization of the optimal load ratio of isolated protocols, we show that the optimal load ratios of (parallel or adaptive) joint protocols are approximately equal to a constant

$$\lambda^* := \left(1 - \sum_{q \in supp(\mathcal{D}_{\boldsymbol{q}})} \mathcal{D}_{\boldsymbol{q}}(q)h(q)\right)^{-1}. \tag{2}$$

where $h(q) := q \log_2 \frac{1}{q} + (1-q) \log_2 \frac{1}{1-q}$ is the entropy function. In other word, $\lambda^*$ is the average channel capacity of the conference review problem. Formally, for any $\delta, \epsilon > 0$, there is no $(\lambda^* - \delta, \epsilon)$-adaptive protocol, but there exists a $(\lambda^* + \delta, \epsilon)$-nonadaptive conference protocol, when the number of papers $n$ is sufficiently large. In other word, the optimal load ratio with and without feedback converges to $\lambda^*$ as $n$ increases. This result contrasts with the isolated setting in Theorem 4.1, where the optimal load ratio increases as the number of papers grows.

**Theorem 4.5.** *Given any conference review problem with $(n, \mathcal{D}_q)$, let $\lambda^*$ defined in Eq. (2) and $V = \max_{q \in supp \mathcal{D}_q} \log_2 \frac{1-q}{q}$. For all $\epsilon > 0$, any $(\lambda, \epsilon)$-adaptive protocol has $\lambda \geq \lambda^* \left(1 - \epsilon - \frac{h(\epsilon)}{n}\right)$.*

*Conversely, for all $\delta > 0$ there exists a parallel $((1 + 2\delta)\lambda^*, P_e)$-conference protocol where error probability $P_e \leq \exp\left(-\frac{2\delta^2}{(1+2\delta)\lambda^* V^2} n\right) + \exp(-\delta n)$.*

To prove Theorem 4.5, we use information theory to show that no protocol can achieve an error probability of $\epsilon$ and load much smaller than $\lambda^*$. The key idea is to first bound the mutual information between the paper quality $\mathbf{z}$ and the reports $\mathbf{y}$. Then we apply Fano's inequality Theorem A.2 to lower bound the error probability. Conversely, we show the existence of joint parallel review protocol with error probability $\epsilon$ and load around $\lambda^*$ using Theorem A.3. The proof structure closely follow the proof of Shannon's noise channel theorem in Polyanskiy & Wu (2014).

The above results show the existence of parallel conference protocols that perform similarly as any adaptive ones for each $n$. This result can be seen as a restatement of Shannon's noisy channel theorem, which shows that the capacity of the channel decides the maximum size of message that a protocol can reliably transmit. Our optimal load ratio $\lambda^*$ corresponds to the average capacity of channel $\otimes_i BSC(q_i)$. Finally, although our proof does not provide an explicit construction, we can leverage existing optimal codes to design near-optimal review protocols, using the connections in Remark 3.1. For example, we may use the low density parity checking code (LDPC) Gallager (1962) to design review protocol where each task $f_i$ is an XOR task with subset of papers $S_i \subseteq [n]$. Additionally, these sets $S_i$ are low density, $|S_i| = o(n)$ as $n \to \infty$ MacKay (1999).

## 5 EXPERIMENTS: TWO-PHASE REVIEW PROTOCOLS

This section empirically demonstrates the potential of adaptive protocols to reduce the load ratio in joint review processes. We focus on the design of two-phase review protocols and their performance compared to parallel baselines under multiple scenarios. We find that two-phase review protocols can achieve better performance with appropriate sets of parameters and assignment strategies.

### 5.1 UNIFORM NOISE LEVELS

In previous sections we discussed joint review protocols with known noise levels. Here we explore the influence of $\lambda$ and $n$ on the performance of two-phase reviews as well as the direct review baselines. Given $\lambda, n$, the design of two-phase reviews (also as two-stage reviews) involves three aspects: the fraction of papers entering the second stage $\eta$, the assignment of workloads in the 2 stages $\lambda_1, \lambda_2$, and the paper selection methods. In our experiment, the number of papers is set to 100, with the same number of reviewers who have a uniform noise level $r = 0.3$. We tested $\eta$ from 0.1 to 0.9. As for the workload assignment, we develop a set of assignment strategies depending on the selection of $\eta$ while keeping the overall $\lambda$ (the ratio of all review tasks over all reviewers) stable.

In the first stage, we aggregate the reviewers' ratings through belief propagation (Liu et al., 2012). We then pick papers for Stage 2 based on the BP scores $y_{BP}$ where a positive large $y_{BP}$ means the paper's expected quality is large and is rated as acceptable and $y_{BP} < -1$ indicates the paper should be rejected. We design two strategies for paper selection:

- The *Ambiguous* strategy picks papers with smallest $|y_{BP}|$ as these papers have higher risks of being misclassified.

- The *Promising* strategy picks papers with highest $y_{BP}$ as most two-phase reviews in reality currently do: the first stage filters out a fraction of the worst papers.

After reviewers in the second stage completed their tasks, we aggregate the opinions of both stages with belief propagation for the final decision.

The total workload for all experimental groups was fixed at $\lambda = 8$. We evaluated performance using **calibration error**, a metric that quantifies the divergence between the predicted distribution of paper qualities and the true underlying distribution. Our results show that optimal performance is achieved with $\eta = 0.4$ when using the *Ambiguous* selection strategy. This optimum occurred under the *Original Strategy* for workload allocation, which sets $\lambda_1 = 4.8$ and $\lambda_2 = 8.0$. The details of this and other allocation strategies are provided in Appendix D.

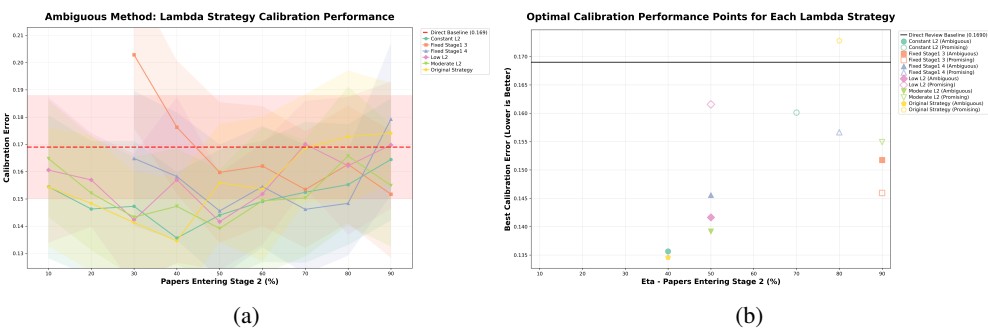

(a)          (b)

Figure 2: (a)Simulations comparing Two-phase *Ambiguous* reviews with direct baselines. Different curves represent diverse workload assignment strategies. (b)The optimal performance is marked by the yellow pentagon, achieved by two-phase *Ambiguous* strategy with $\eta = 0.4$.

Using the optimal set of parameters identified in the previous experiment, we next investigate the influence of the total workload, $\lambda$, on review performance. We evaluate performance using two metrics: calibration error and accuracy, where accuracy is defined as the fraction of correctly classified papers: $\text{Acc} = \frac{\#\text{Correctly Classified Papers}}{\#\text{All Papers}}$ Our empirical results show a clear positive trend: as $\lambda$ increases, performance measured by both metrics consistently improves.

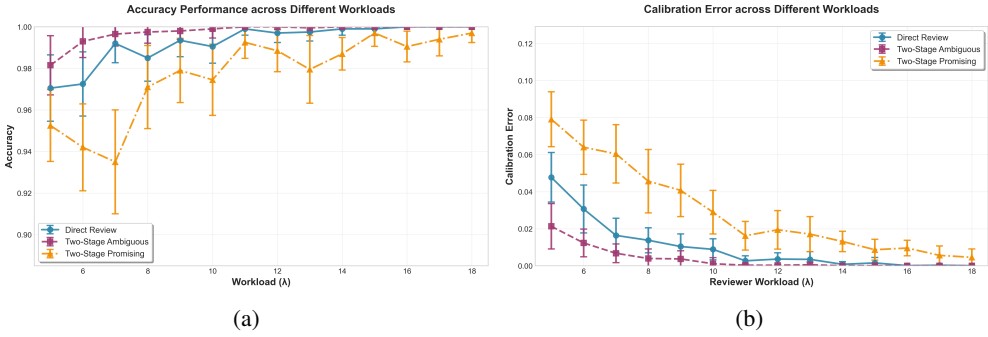

(a)          (b)

Figure 3: Simulation results illustrating better review performances with higher workloads. As $\lambda$ increases, the accuracy of all methods exhibits a rising trend and the calibration error decreases. The *Ambiguous* strategy dominates the other 2 strategies for every selection of $\lambda$.

The recent surge in submissions to academic conferences has raised significant concerns about maintaining review quality at scale. Motivated by this trend, we investigate how review performance is affected by an increasing number of papers, $n$. In these simulations, we scale the number of reviewers to match the number of papers, maintaining a one-to-one ratio. The total workload is held constant at $\lambda = 8$, and we employ the optimal parameter set for $\eta$, $\lambda_1$, and $\lambda_2$ as determined in the preceding experiment.

Our results, detailed in Appendix E, reveal a decline in the accuracy of the two-phase review process as the number of papers increases. This finding highlights the negative impact that a larger scale has on overall review performance.

## 5.2 Unknown Noise Levels

We now extend our analysis to a more realistic setting where the specific noise level of each reviewer is unknown. We adopt the well-known spammer-hammer framework Karger et al. (2011). The reviewer pool is assumed to consist of two distinct groups: 50% are expert "hammers" with a low noise level ($r_1 = 0.1$), and the remaining 50% are non-expert "spammers" with a high noise level ($r_2 = 0.5$). While individual reviewer types are unknown, we assume the prior distribution of these types is accessible to the belief propagation algorithm. Our objective is twofold: first, to determine whether the two-phase review process remains beneficial, and second, to identify the optimal configuration of $\eta$, $\lambda_1$, and $\lambda_2$.

Using accuracy as the evaluation metric, our results show that the *Ambiguous* strategy outperforms direct reviews across most values of $\eta$. Furthermore, the optimal performance is achieved at $\eta = 0.3$ when employing the *Ambiguous* strategy.

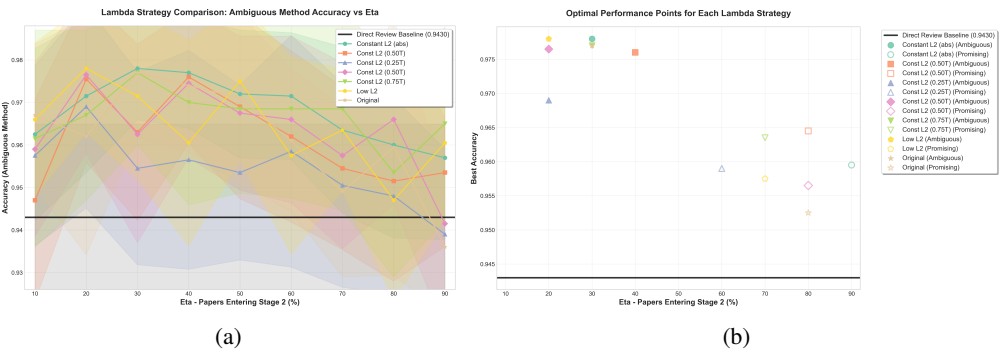

(a)                                                                 (b)

Figure 4: (a)Simulation results showing higher accuracies of the *Ambiguous* strategy compared to direct review baselines. (b)The green dot marks the optimal performance of two-phase reviews, achieved by the *Ambiguous* strategy with $\eta = 0.3$.

## 6 Conclusion and Discussion

We initiate the conference protocol design problem. By viewing peer review through an information theoretic lens, we prove that in an isolated protocol, direct review is Blackwell optimal and any protocol needs a load ratio $\Theta(\ln n/\epsilon)$, and a joint protocol can achieve a constant load ratio. We further empirically explore two-phase review protocols and show that focusing on ambiguous papers can outperform single-stage baselines.

We assume an agent's noise level is *task-agnostic*—it does not vary with the complexity of the assigned task. This is plausible for protocols in which each task is a single-paper judgment (e.g., direct review, naive meta-review, and two-phase schemes where each task still evaluates a single paper). However, if the noise level in adaptive protocols is higher than in parallel ones, by Proposition 4.3, our results in Lemma 4.2 and Theorem 4.5 for both isolated and joint protocols still hold. Additionally, research on paper assignment problems has considered scenarios where reviewers have different areas of expertise, leading to various noise levels when evaluating different papers. Investigating the interplay between expertise-based assignment and task-dependent noise could be an exciting future work.

Finally, our analysis can be extended beyond the assumptions of binary paper quality and reports and the symmetric noise model. While the Blackwell dominance result in Lemma 4.2 may no longer hold, Theorem 4.5 can naturally accommodate general report and paper quality states by leveraging coding theory for non-binary settings. As a joint protocol is equivalent to an isolated setting with one paper, the above argument implies the direct review protocol is also optimal in the non-binary case. Our results also generalize beyond the uniform prior assumption using techniques from joint source channel coding. Similarly, different accuracy guarantees—such as minimizing the average number of errors, which is well-studied in coding theory—can also be explored within this framework.

## ETHICS STATEMENT

This paper tends to investigate whether complex protocols can reduce the review load ratio in conference peer review processes. We believe our work will have a potentially positive impact on conference reviews and other crowd sourcing tasks.

## REPRODUCIBILITY STATEMENT

All experiments in this paper are generally reproducible. We carefully described the experiment details and parameter configurations in Section 5 and provided codes along with documentation in the supplementary materials.

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

## A  PRELIMINARY

The first half of Appendix A.1 provides basic nomenclature for coding theory and highlights an interesting analogy between conference problems and coding problems, as discussed in Remark 3.1. The second half defines essential notations from information theory needed for our proofs.

### A.1  CODING THEORY AND INFORMATION THEORY

A *channel* consists of an input space $\mathcal{X}^m$, an output space $\mathcal{Y}^m$, and a transition matrix $P_{\mathbf{y}|\mathbf{x}} = \boldsymbol{Q}$. For any input $\boldsymbol{x} \in \mathcal{X}^m$ the channel distorts $\boldsymbol{x}$ to a random $\mathbf{y} \in \mathcal{Y}^m$ according to an independent sample drawn according to $\boldsymbol{Q}(\cdot \mid \boldsymbol{x})$ denoted as $\boldsymbol{x} \xrightarrow{\boldsymbol{Q}} \mathbf{y}$. For instance, given $m = 1$, $\mathcal{X} = \mathcal{Y} = \{-1, 1\}$, and $q \in [0, 1]$, the *binary symmetric channel* $BSC(q)$ takes $x$ and outputs $\mathbf{y} \xleftarrow{q} x$. This paper will mostly consider composition of binary symmetric channels with transition kernel $\boldsymbol{Q} = \otimes_{i=1}^m BSC(q_i)$ where each $\mathbf{y}_i \xleftarrow{q} x_i$ independently and $\otimes$ denotes the direct product of those transition matrices.

Given a channel $(\mathcal{X}^m, \mathcal{Y}^m, \boldsymbol{Q})$, an $N$-code for $\boldsymbol{Q}$ is an encoder/decoder pair $(\boldsymbol{f}, g)$ of (randomized) functions

- encoder $\boldsymbol{f} = (f_1, \ldots, f_m) : [N] \to \mathcal{X}^m$
- decoder $g : \mathcal{Y}^m \to [N]$.

The underlying probability space for channel coding problem is

$$\mathbf{w} \xrightarrow{f} \mathbf{x} \xrightarrow{\boldsymbol{Q}} \mathbf{y} \xrightarrow{g} \hat{\mathbf{w}}$$

where $\mathbf{w}$ is the original message, $\mathbf{x}$ is channel input, $\mathbf{y}$ is channel output, and $\hat{\mathbf{w}}$ is the decoded message. The *average error probability* is $P_e := \Pr[\mathbf{w} \neq \hat{\mathbf{w}}]$ where $\mathbf{w}$ is uniformly sampled from $[N]$.

A code $(\boldsymbol{f}, g)$ is an $(N, \epsilon)$-code for $\boldsymbol{Q}$ if the message space is $[N]$ and $P_e \leq \epsilon$, and $(m, N, \epsilon)$-code if the input space of the channel is product space $\mathcal{X}^m$.

This equivalence also extends to adaptive protocols and codes with feedback. An $(m, N, \epsilon)$-code with feedback is specified by the encoder-decoder pair $(\boldsymbol{f}, g)$ as follows:

- Encoder:

$$f_1 : [N] \to \mathcal{X}$$
$$f_2 : [N] \times \mathcal{Y} \to \mathcal{X}$$
$$\ldots$$
$$f_m : [N] : \times \mathcal{Y}^{m-1} \to \mathcal{X}$$

- Decoder $g : \mathcal{Y}^m \to [N]$

such that error probability $P_e \leq \epsilon$. Here the symbol transmitted at round $i$ depends on both the message w and the history of received symbols $\mathrm{x}_i = f_i(m, \mathbf{y}_{1:i-1})$. Again, a code with feedback corresponds to an adaptive review and inference protocol.

**Primer on information theory** After establish the analogy between coding theory and conference problem, we introduce some necessary notations and results for our proofs. These results are standard and can be found in textbooks such as Thomas & Joy (2006); Polyanskiy & Wu (2014). Given a joint distribution $P_{\mathrm{x,y}}$ for $(\mathrm{x}, \mathrm{y}) \in \mathcal{X} \times \mathcal{Y}$, the *information density* on $(x, y) \in \mathcal{X} \times \mathbf{Y}$ is

$$J_{P_{\mathrm{x,y}}}(x; y) := \log \frac{P_{\mathrm{x,y}}(x, y)}{P_{\mathrm{x}}(x) P_{\mathrm{y}}(y)}$$

where $P_{\mathrm{x}}$ and $P_{\mathrm{y}}$ are marginal distributions. We will drop subscript when the joint distribution is clear in context. The (Shannon) *mutual information* is

$$I(\mathrm{x}; \mathrm{y}) := \mathbb{E}[J(\mathrm{x}; \mathrm{y})].$$

Equivalently, given two distributions $P$ and $Q$ on a discrete space $\Omega$, the KL divergence is $D_{KL}(P||Q) := \sum_{w \in \Omega} P(w) \log \frac{P(w)}{Q(w)}$, and the mutual information is $I(\mathrm{x}; \mathrm{y}) = D_{KL}(P_{\mathrm{x,y}}||P_{\mathrm{x}} \otimes P_{\mathrm{y}})$ where $P_{\mathrm{x}} \otimes P_{\mathrm{y}}$ is the direct product the marginal distributions. For brevity, we also write them as $P_{\mathrm{x}}P_{\mathrm{y}}$.

Additionally, given a transition matrix $P_{\mathrm{y|x}} = \boldsymbol{Q}$, the *capacity* of $\boldsymbol{Q}$ is

$$C := \sup_{P_{\mathrm{x}}} I(\mathrm{x}; \mathrm{y}) \tag{3}$$

which is the maximum mutual information under all possible marginal distribution of x, $P_{\mathrm{x}}$, and $\mathrm{y} \xleftarrow{\boldsymbol{Q}} \mathrm{x}$.

The following lemma gives basic properties of mutual information and capacity. Let entropy function be $h(q) := q \log_2 \frac{1}{q} + (1 - q) \log_2 \frac{1}{1-q}$ for any $q \in [0, 1]$ where $\log_2$ denotes the logarithm with base 2.

**Lemma A.1.** *If* $\boldsymbol{Q} = \otimes_i \boldsymbol{Q}_i$, *for all* $m \in \mathbb{N}_{>0}$

$$\sup_{P_{\mathbf{x}^m}} I(\mathbf{x}^m; \mathbf{y}^m) = \sum_{i=1}^{m} \sup_{P_{\mathrm{x}_i}} I(\mathrm{x}_i; \mathrm{y}_i).$$

*For binary symmetric channel* $BSC(q)$

$$\sup_{P_{\mathrm{x}}} I(\mathrm{x}; \mathrm{y}) = 1 - h(q)$$

*and the optimal of Eq.* (3) *happens when* x *is uniformly distributed.*

Note that $h(q) \in [0, 1]$ as we are using 2 as the base, and $1 - h(q)$ achieve minimum when $q = 1/2$ which corresponds to large noise level setting.

The following inequality lower bounds the error probability through mutual information.

**Theorem A.2** (Fano's inequality). *Let* $(\mathrm{x}, \mathrm{y})$ *be a random variable on* $[N]^2$ *with joint distribution* $P_{\mathrm{x,y}}$, *and let* $Q_{\mathrm{x,y}} = P_{\mathrm{x}}P_{\mathrm{y}}$ *be the product of marginal distributions*

$$I(\mathrm{x}; \mathrm{y}) \geq P_{\mathrm{x,y}}[\mathrm{x} = \mathrm{y}] \log_2 \frac{1}{Q_{\mathrm{x,y}}[\mathrm{x} = \mathrm{y}]} - h(P_{\mathrm{x,y}}[\mathrm{x} = \mathrm{y}])$$

*Additionally if* $P_{\mathrm{x}}$ *or* $P_{\mathrm{y}}$ *is uniform distribution*

$$I(\mathrm{x}; \mathrm{y}) \geq (1 - P_{\mathrm{x,y}}[\mathrm{x} \neq \mathrm{y}]) \log_2 N - h(P_{\mathrm{x,y}}[\mathrm{x} \neq \mathrm{y}])$$

We introduce Shannon's achievability bound Theorem A.3, which shows the existence of a code whose error probability is bounded by the tail probability of the information density $J$. Because the expectation of $J$ is the mutual information, we can have a small error probability when the message size $N$ is of the order $2^C$, establishing one side of the renowned Shannon's Noisy Channel Theorem.

**Theorem A.3** (Shannon's achievability bound). *Given* $P_{\mathrm{y|x}} = \boldsymbol{Q}$, *for any* $P_{\mathrm{x}}$, $\tau > 0$, *there exists a* $(N, \epsilon)$-*code without feedback such that*

$$\epsilon \leq \Pr[J(\mathrm{x}; \mathrm{y}) \leq \log_2 N + \tau] + e^{-\tau}.$$

## A.2 BLACKWELL DOMINANCE

Given a finite state of the world $\Omega$, an experiment $(\mathcal{S}, (P_\omega)_{\omega \in \Omega})$ consists of a signal space $\mathcal{S}$, and a collection of probability measures $(P_\omega)_{\omega \in \Omega}$ with the interpretation that $P_\omega(y)$ is the probability of observing $y \in \mathcal{Y}$ in state $\omega \in \Omega$. In our setting, paper quality corresponds to the state, reports serve as signals, and each review protocol represents a different experiment.

**Definition A.4** (Blackwell et al. (1951)). *Given two experiments $P = (\mathcal{S}, (P_\omega)_{\omega \in \Omega})$ and $Q = (\mathcal{S}, (Q_\omega)_{\omega \in \Omega})$, we say $P$ Blackwell dominates $Q$ if there exists a measurable kernel (known as garbling) $\pi : \mathcal{S} \to \Delta\mathcal{S}$ where $\Delta\mathcal{S}$ is the set of probability measures over $\mathcal{S}$ such that for any $\omega$ and $s \in \mathcal{S}$*

$$Q_\omega(s) = \int \pi(s', s) dP_\omega(s')$$

*In other words, there is a possibly random simulation $h : \mathcal{S} \to \mathcal{S}$ so that for all $\omega$, if $S'$ is distributed according to $P_\omega$ then $S = h(S')$ is distributed according to $Q_\omega$.*

# B PROOFS AND DETAILS FOR SECTION 3

Because the optimal inference protocol can be MLE, we will term conference protocol and review protocol interchangeably. For example, a direct protocol refers to a conference protocol that uses a direct review protocol along with the MLE inference protocol. The following proposition further shows that we only need to focus on deterministic review protocols.

**Proposition B.1.** *Given any conference problem $(n, \mathcal{D}_q)$, there exists a deterministic conference protocol $(f, g)$ that minimizes error $P_e$.*

*Proof of Proposition B.1.* Any randomized review protocol $f$ with an inference protocol $g$ can be represented as a deterministic review protocol $f_r$ with external randomness $\mathrm{r} = r$. Let $P_e(r)$ be the error probability when the $\mathrm{r} = r$ under the inference protocol $g$. Because $P_e = \Pr[\mathbf{z} \neq \hat{\mathbf{z}}] = \mathbb{E}_{\mathrm{r}}[\Pr[\mathbf{z} \neq \hat{\mathbf{z}}] \mid \mathrm{r}] = \mathbb{E}_{\mathrm{r}}[P_e(\mathrm{r})]$, there exists $r^*$ so that $P_e(r^*) \leq P_e$. Therefore, a deterministic review protocol $f_{r^*}$ with an MLE inference protocol must have a smaller or equal error probability. $\square$

# C PROOFS AND DETAILS IN SECTION 4

## C.1 BLACKWELL DOMINANCE OF DIRECT REVIEW PROTOCOLS

We prove Lemma 4.2 by direct construction of a direct protocol that Blackwell dominates $f$. Since $f$ is an isolated review protocol, each task is associated with at most one paper. We construct $f'$ as a direct review protocol by assigning each task to directly evaluate its associated paper. Formally, let $\Gamma_j \subseteq [m]$ be the set of tasks assigned to paper $j$. Because $f$ is isolated, the sets $\Gamma_j$ are disjoint. Therefore, we can define a direct protocol $f'$ via a mapping $\sigma : [m] \to [n]$ such that $\sigma(i) = j$ and task $i$ reviews paper $j$ for all $i \in \Gamma_j$. Because the distributions of reports across different $\Gamma_j$ are mutually independent, it is sufficient to show Blackwell dominance under the single-paper setting $n = 1$ as shown in Lemma C.1, to complete the proof of Lemma 4.2.

**Lemma C.1.** *Given any conference review problem $(1, \mathcal{D}_q)$, the direct review protocol $f'$ Blackwell dominates any review protocol $f$ with the same review load ratio.*

*Proof of Lemma C.1.* We construct a function from direct review protocol with $\mathbf{y}' = \mathbf{y}'$ where $\mathrm{y}'_i \xleftarrow{q_i} f'_i(z) = z$ for all $i$ to $\tilde{\mathbf{y}}$ to simulate the reports from isolated review protocol $\mathbf{y}$.

Our function iteratively construct $\tilde{\mathbf{y}}$ from $i = 1$ to $i = m$. Given $\mathbf{y}' = \mathbf{y}'$ and $f$, there are four cases for each $i$.

1. We set $\tilde{\mathrm{y}}_i = \mathrm{y}'_i$, if $f_i(z, \tilde{\mathbf{y}}_{1:i-1}) = z$ for $z \in \{-1, 1\}$.

2. We set $\tilde{\mathrm{y}}_i = -\mathrm{y}'_i$, if $f_i(z, \tilde{\mathbf{y}}_{1:i-1}) = -z$ for $z \in \{-1, 1\}$.

3. We set $\tilde{\mathrm{y}}_i \xleftarrow{q_i} 1$, if $f_i(z, \tilde{\mathbf{y}}_{1:i-1}) = 1$ for $z \in \{-1, 1\}$.

4. We set $\tilde{y}_i \xleftarrow{q_i} -1$, if $f_i(z, \tilde{\mathbf{y}}_{1:i-1}) = -1$ for $z \in \{-1, 1\}$.

Now we use induction to show that $\tilde{\mathbf{y}}$ and the reports from the isolated review protocol $\mathbf{y}$ are equal in distribution for all $z \in \{-1, 1\}$. This shows that $\mathbf{y}'$ Blackwell dominates $\mathbf{y}$. Consider the base case, $i = 1$. For the first case, $f_1(z) = z$ for all $z \in \{-1, 1\}$, $y_1$ is also a direct review task $y_1 \xleftarrow{q_1} z$, and by definition $\tilde{y}_1 = y_1 \xleftarrow{q_1} z$. For the second case, $y_1$ is the flipped direct review, because $y_1 \xleftarrow{q_1} -z$ is equivalent to $-y_1 \xleftarrow{q_1} z$. The third and fourth hold analogously. Therefore, the random variables are equal in distribution,

$$y_1|z \overset{d}{=} \tilde{y}_1|z$$

for all $z$. For inductive steps, suppose $\mathbf{y}_{1:i-1}|z \overset{d}{=} \tilde{\mathbf{y}}_{1:i-1}|z$. By the above arguments,

$$y_i|z, \mathbf{y}_{1:i-1} = \boldsymbol{y}_{1:i-1} \overset{d}{=} \tilde{y}_i|z, \tilde{\mathbf{y}}_{1:i-1} = \boldsymbol{y}_{1:i-1}$$

for all $z$ and any partial realization $\tilde{\mathbf{y}}_{1:i-1} = \mathbf{y}_{1:i-1} = \boldsymbol{y}_{1:i-1}$. Therefore, $\mathbf{y}_{1:i}|z \overset{d}{=} \tilde{\mathbf{y}}_{1:i}|z$ which completes the proof. $\qquad\square$

**Proposition C.2.** *Given any conference review problem $(n, \mathcal{D}_{\boldsymbol{q}})$ and $m$, successive review protocols with load ratio $\frac{m}{n}$ and direct review protocols with load ratio $\frac{m}{n}$ are Blackwell equivalent.*

*Proof of Proposition C.2.* Similar to Lemma 4.2, as the protocol is isolated, we only need to consider single paper $n = 1$, $f_1$ is a direct review and $f_{i+1}$ checks if the $i$-th report is correct. Hence, $x_1 = z$, and

$$f_{i+1}(z, y_1, \ldots, y_i) = f_i(z, y_1, \ldots, y_{i-1}) \oplus y_i = z \oplus y_1 \oplus \cdots \oplus y_i. \tag{4}$$

Given reports $y_1, \ldots, y_m$ for the above successive protocol, we set $\tilde{y}_i = \oplus_{j=1}^i y_j$ for all $i$. By Eq. (4), $\tilde{y}_i \xleftarrow{q_i} z \oplus (\oplus_{j=1}^i y_j) \oplus (\oplus_{j=1}^i y_j) = z$, so the distribution of $\tilde{\boldsymbol{y}} = (\tilde{y}_1, \ldots, \tilde{y}_m)$ is identical to the distribution of $m$ direct review protocol. For the other direction, given reports $y'_1, \ldots, y'_m$ from the direct review protocol, by above derivation, $(\oplus_{j=1}^i y'_j)_{i=1,\ldots,m}$ has the same distribution as the successive protocol. $\qquad\square$

Note that in the proof of Proposition C.2, our reduction does not require the knowledge of histogram of noise level $\mathcal{D}_{\boldsymbol{q}}$, and is applicable in the unknown noise levels.

*Proof of Lemma 4.4.* Given $\lambda$ and $m = \lambda n$, we provide upper and lower bounds of any direct review protocols. Let $Bin(k, p)$ denote Binomial random variables with parameter $k \in \mathbb{N}$ and $p \in [0, 1]$ so that $\Pr[Bin(k, p) = l] = \binom{k}{l} p^l (1-p)^{k-l}$ for all $0 \leq l \leq k$.

First note that if the noise level is homogeneous, we can specify a direct review by the number of review each paper $j$ gets $a_j$ and $\sum_j a_j = m$. Additionally, the optimal inference protocol is the majority vote, and the error probability of a single paper is

$$\Pr[\hat{z}_j \neq z_j] = \Pr\left[Bin(a_j, q) \geq \left\lfloor \frac{a_j + 1}{2} \right\rfloor\right]$$

where we ignore tie breaking term. Let $a_j = 2k$ be even. By Klar (2000) we can bound the above using the probability mass function $\Pr[Bin(2k, q) = k]$,

$$1 \leq \frac{\Pr[Bin(2k, q) \geq k]}{\Pr[Bin(2k, q) = k]} \leq \frac{(k+1)(1-q)}{k+1-(2k+1)q} \leq \frac{1-q}{1-2q} \tag{5}$$

By Stirling formula $\frac{4^k}{2\sqrt{\pi k}} < \binom{2k}{k} < \frac{4^k}{\sqrt{\pi k}}$, (https://math.stackexchange.com/users/14812/robert-william hanks)

$$\frac{\Pr[Bin(2k, q) = k]}{\frac{4^k}{\sqrt{\pi k}} q^k (1-q)^k} = \frac{\binom{2k}{k}}{\frac{4^k}{\sqrt{\pi k}}} \in [1/2, 1] \tag{6}$$

Combining Eqs. (5) and (6), we have $\frac{1}{2} \leq \frac{\Pr[Bin(2k,q)\geq k]}{\frac{4^k}{\sqrt{\pi k}}q^k(1-q)^k} \leq \frac{1-q}{1-2q}$. The error probability of a single paper with $2k$ direct reviews can be approximated by a convex function

$$p(2k, q) := \frac{1}{\sqrt{\pi k}}(4q(1-q))^k.$$

Now we give a lower bound on error probability $P_e$. Because the error probability of each paper is independent, the error probability is $P_e \geq 1 - \prod_{j=1}^{n}(1 - \frac{1}{2}p(a_j, q))$. By AM-GM inequality, $\prod_{j=1}^{n}(1 - \frac{1}{2}p(a_j, q)) \leq (1 - \frac{1}{n}\sum_j \frac{1}{2}p(a_j, q))^n$. Because $p(a_j, q)$ is convex, $\frac{1}{n}\sum_j p(a_j, q) \geq p(\frac{1}{n}\sum_j a_j, q) = p(\frac{m}{n}, q) = p(\lambda, q)$. Therefore,

$$P_e \geq 1 - (1 - \frac{1}{n}\sum_j \frac{1}{2}p(a_j, q))^n \geq 1 - \left(1 - \frac{1}{2}p(\lambda, q)\right)^n.$$

Therefore, if $P_e \leq \epsilon$,

$$(1-\epsilon)^{1/n} \leq 1 - \frac{1}{2}p(\lambda, q)$$

$$\frac{1}{2}p(\lambda, q) = O(\epsilon/n) \qquad \text{(Taylor expansion)}$$

$$\frac{1}{\sqrt{\pi\lambda/2}}(4q(1-q))^{\lambda/2} = O(\epsilon/n)$$

$$\lambda = \Omega(\ln \frac{n}{\epsilon}) \qquad (4q(1-q) < 1)$$

Similar, by assigning the same number of direct review to each paper, we have upper bound

$$P_e \leq 1 - \left(1 - \frac{1-q}{1-2q}p(\lambda, q)\right)^n,$$

and $\lambda = \Omega(\ln \frac{n}{\epsilon})$ is sufficient for $P_e \leq \epsilon$. $\qquad\square$

*Proof of Theorem 4.1.* First we consider direct protocols on homogeneous noise level where every task has noise level $q$. Lemma 4.4 shows that under the homogeneous setting a symmetric direct protocol is optimal and provides an upper bound of the load ratio. Second, it directly provide a lower bound on any possible direct protocol.

For upper bound in the heterogeneous setting, we take the largest noise level $\overline{q} = \max\{q \in supp(\mathcal{D}_q)\}$, and compute the load ratio of the symmetric direct protocol under a homogeneous setting $(n, \overline{q}\mathbf{1})$. By Proposition 4.3, there exist a (direct) protocol under $(n, \mathcal{D}_q)$ that Blackwell dominates the symmetric direct protocol and thus has the same load ratio $O(\ln n/\epsilon)$ with error probability $P_e \leq \epsilon$.

For lower bound, we take the smallest noise level $\underline{q} = \min\{q \in supp(\mathcal{D}_q)\}$. Since by Lemma 4.2 and Proposition 4.3 the reports from any adaptive protocols are Blackwell dominated by the reports from some direct protocol under $(n, \underline{q}\mathbf{1})$. Additionally, any direct protocol under $(n, \underline{q}\mathbf{1})$ has load ratio at least $\Omega(\ln n/\epsilon)$ by Lemma 4.4 which completes the proof. $\qquad\square$

## C.2 Joint review protocol

*Proof of Theorem 4.5.* For the first part, we lower bound the load ratio for error probability $\epsilon$. For any adaptive protocol with $m = \lambda n$ review tasks, we have

$$
\begin{aligned}
-h(\epsilon) + (1 - \epsilon) \log_2 2^n \leq &I(\mathbf{z}; \hat{\mathbf{z}}) && \text{(by Fano's inequality Theorem A.2)} \\
\leq &I(\mathbf{z}; \mathbf{y}) && \text{(data processing inequality)} \\
= &\sum_{i=1}^{m} I(\mathbf{z}; \mathbf{y}_i \mid \mathbf{y}^{i-1}) && \text{(Chain rule)} \\
\leq &\sum_{i=1}^{m} I(\mathbf{z}, \mathbf{y}^{i-1}; \mathbf{y}_i) && (I(\mathbf{z}; \mathbf{y}_i \mid \mathbf{y}^{i-1}) = I(\mathbf{z}, \mathbf{y}^{i-1}; \mathbf{y}_i) - I(\mathbf{y}^{i-1}; \mathbf{y}_i)) \\
\leq &\sum_{i=1}^{m} I(\mathbf{x}_i; \mathbf{y}_i) && \text{(data processing inequality)} \\
\leq &\sum_{i=1}^{m} (1 - h(q_i)) = \frac{m}{\lambda^*} && \text{(Lemma A.1)}
\end{aligned}
$$

Therefore,

$$
\lambda = \frac{m}{n} \geq \lambda^* \left( (1 - \epsilon) - \frac{h(\epsilon)}{n} \right).
$$

For the second part, consider $(\mathbf{x}, \mathbf{y})$ where $\mathbf{x}_i \in \{-1, 1\}$ is uniformly distributed and $\mathbf{y}_i \xleftarrow{q_i} \mathbf{x}_i$ independently for all $i \in [m]$. For any $\delta > 0$, $\lambda = (1 + 2\delta)\lambda^* > \lambda^* > 1$, and $\tau = \delta n$, by Theorem A.3 there exists $(\lambda n, 2^n, \epsilon)$-code such that

$$
\epsilon \leq \Pr[J(\mathbf{x}; \mathbf{y}) \leq \log_2 2^n + \delta n] + e^{-\delta n}.
$$

Because each coordinate of $(\mathbf{x}, \mathbf{y})$ is independent, we can define random variables $\mathbf{z}_i := J(\mathbf{x}_i, \mathbf{y}_i)$ with $i \in [m]$ which are mutually independent with expectation $\mathbb{E}\mathbf{z}_i = I(\mathbf{x}_i; \mathbf{y}_i) = \sup_{P_{\mathbf{x}_i}} I(\mathbf{x}_i; \mathbf{y}_i) = 1 - h(q_i)$ by Lemma A.1 and $J(\boldsymbol{x}; \boldsymbol{y}) = \sum_i \mathbf{z}_i$. Hence,

$$
\mathbb{E}[J(\mathbf{x}; \mathbf{y})] = \sum_{i=1}^{m} \mathbb{E}[\mathbf{z}_i] = \sum_{i=1}^{m} 1 - h(q_i) = \frac{m}{\lambda^*} = (1 + 2\delta)n > n.
$$

Because $\log_2 2q_i \leq \mathbf{z}_i \leq \log_2 2(1 - q_i)$, by Hoeffding's inequality,

$$
\Pr[J(\mathbf{x}; \mathbf{y}) \leq \log_2 2^n + \delta \lambda^* n] = \Pr\left[ \sum_i (\mathbf{z}_i - \mathbb{E}\mathbf{z}_i) \leq -\delta n \right] \leq \exp\left( -\frac{2\delta^2 n^2}{mV^2} \right) = \exp\left( -\frac{2\delta^2}{\lambda V^2} n \right)
$$

which completes the proof. $\qquad\square$

# D  ASSIGNMENT STRATEGY

In this section we introduce assignment strategies used in Section 5, where $\lambda_{\text{tot}}$ is the overall $\lambda$, i.e., $\lambda_{\text{tot}} = \lambda_1(\eta) + \eta \lambda_2(\eta)$.

| Strategy | $\lambda_1(\eta)$ | $\lambda_2(\eta)$ |
|---|---|---|
| Original Strategy | $\lambda_{\text{tot}}(1 - \eta)$ | $\lambda_{\text{tot}}$ |
| Constant L2 | $\lambda_{\text{tot}} - 6\eta$ | 6 |
| Moderate L2 | $\lambda_{\text{tot}} - 8\eta$ | 8 |
| Low L2 | $\lambda_{\text{tot}} - 3\eta$ | 3 |
| Fixed Stage1-4 | 4 | $(\lambda_{\text{tot}} - 4)/\eta$ |
| Fixed Stage1-3 | 3 | $(\lambda_{\text{tot}} - 3)/\eta$ |

# E  SUPPLEMENTARY RESULTS

In this section we display the supplementary results for Section 5.1.

918
919
920
921
922
923
924
925
926
927
928
929
930
931
932
933
934
935
936
937
938
939
940
941
942
943
944
945
946
947
948
949
950
951
952
953

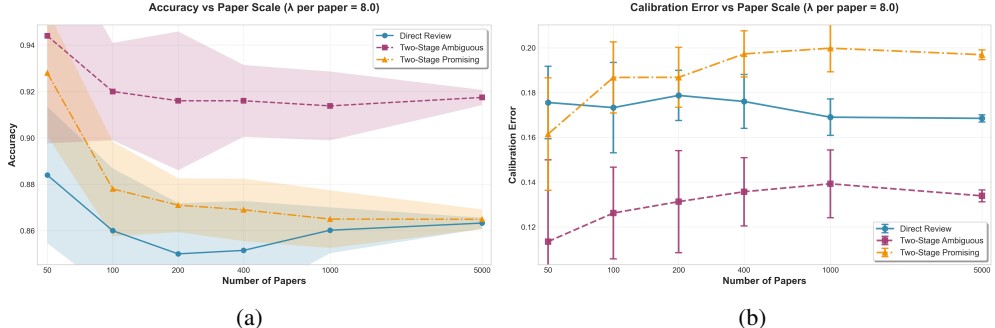

(a)                                                  (b)

Figure 5: Simulation results illustrating the influence of paper numbers on review performances. $n$ ranges from 50 to 5000. As $n$ increases, the accuracy of two-phase reviews shows a downward trend and the calibration error increases. The negative effect is possibly mitigated by the belief propagation based on a larger reviewer population. The *Ambiguous* strategy dominates the other 2 strategies for every selection of $n$.
