# OpenReview forum: "From Crowds to Codes: Minimizing Review Burden in Conference Review Protocols"
_ICLR.cc/2026/Conference — ICLR 2026 Conference Withdrawn Submission_

### Official Review · Reviewer_uqKm · 2025-10-27

**Soundness:** 1
**Presentation:** 1
**Contribution:** 2
**Rating:** 2
**Confidence:** 1

**Summary:**

This paper studies the reviewer load ratio (the number of tasks assigned to each reviewer) across different types of review processes using a mathematical framework. It compares review processes such as ICLR's three independent reviews with two-phased reviews like those used by AAAI. The authors define new attributes (isolation, joint, non-adaptive, adaptive) to characterize peer review processes. The paper claims that selecting borderline papers for second-phase review can reduce reviewer workload.

**Strengths:**

- The paper tackles a relevant problem of reviewer workload management in peer review systems, which is of practical importance to the community.
- The attempt to formalize different review processes through a mathematical framework is a potentially valuable contribution to the field.

**Weaknesses:**

- The reviewer spent an hour on four consecutive days attempting to understand what the paper is trying to convey. I also consulted a colleague to see if they could understand the paper's main message; their assessment matched mine. Despite careful reading, I found the core technical contribution unclear and the writing extremely and unnecessarily complex, with a confusing structure that significantly impacted my ability to assess the work's validity and significance.
- The writing is unnecessarily complex; the paper reads as though the authors are trying to impress rather than communicate. The reader must constantly move back and forth through the paper, as concepts introduced early are defined only later (e.g., $D_q$). Additionally, there are numerous non-intuitive concepts (isolation, joint, non-adaptive, adaptive), which are sometimes used together and sometimes separately, making it confusing for readers to grasp their meanings.
- The paper requires major reformatting and restructuring of its presentation.
- The notation is very confusing. For instance, while boldface for vectors is standard, mixing it with non-bold notation for random variables creates confusion (Line 170). $D_q$ is introduced on Line 172, but $q$ is not defined until Line 187.

**Questions:**

- Is the ICLR review process a meta-protocol or a direct protocol? It would be helpful if the authors could provide real-world examples of each protocol type. Figure 1 is difficult to understand.
- Please use proper citation commands (i.e., \citep{} or \citet{}). The current citation format reduces the readability of the paper.
- Why do the authors use the term "agents" when they could simply refer to them as "reviewers"?
- Is the term "Blackwell" a typo in Line 92?

---

### Official Review · Reviewer_PcGk · 2025-10-30

**Soundness:** 3
**Presentation:** 2
**Contribution:** 2
**Rating:** 4
**Confidence:** 3

**Summary:**

This paper studies conference peer review as an information-theoretic coding problem and asks what minimal reviewer load (number of tasks per paper) is needed to achieve a target decision accuracy. It classifies review protocols along two axes—(i) isolation vs. joint tasks and (ii) parallel vs. adaptive execution, and analyzes each class under an explicit stochastic noise model for reviewer responses.

**Strengths:**

S1: The problem studied in this paper is urgent and well-motivated.

S2: Some observations and conclusions are interesting. Particularly, it shows that joint/comparative tasks can dramatically cut review burden and that parallel (non-adaptive) joint protocols suffice asymptotically.

S3: It separates protocols by isolation/adaptivity, giving organizers a clean design space and shared language.

S4: Definitions, lemmas, and theorems flow logically, making theoretical works in this paper not hard to track.

**Weaknesses:**

W1: The paper assumes i.i.d., task-agnostic reviewer noise, meaning that all reviewers share the same independent error rate regardless of paper topic, complexity, or review round. While this assumption simplifies analysis, it limits realism. In actual conference settings, reviewer performance depends on expertise alignment, workload, and calibration drift. For example, a reviewer who is consistently lenient or harsh across multiple papers introduces cross-task correlations that cannot be captured by an i.i.d. model.

W2: The paper provides limited discussion of strategic or adversarial reviewer behavior, even though real conference review settings are not purely stochastic. The current model assumes honest, independent reviewer responses drawn from a fixed noise distribution, but in practice, reviewers may game or coordinate, e.g., exaggerating scores for favored submissions or colluding within small groups. It is more interesting is the authors can have further discussion about their model in these scenarios.

W3: The paper assumes that each paper’s latent quality $z_j$ is binary ($z_j \in {-1, 1}$), representing a "good" or "bad" paper. While this abstraction simplifies the analysis, it may oversimplify real peer-review settings where paper quality is typically rated on an ordinal or continuous scale (for example, 1–10 scores or multi-criteria rubrics). It is better to consider/discuss whether the main results (such as error bounds and optimality of protocols) could extend to multi-level quality models.

W4: Figure 1 effectively illustrates the overall review-as-communication framework but lacks clear labeling and explanation of key variables. The meanings of x, y, z must be inferred from the text in the later section.

W5: The paper introduces “BP” (e.g., y_BP in Section 5.1) when describing how reviewer ratings are aggregated, but it never explicitly defines what BP stands for. Based on the reference to Liu et al. (2012), the reviewer infers that BP refers to Belief Propagation, but this should be stated clearly when the term first appears.

**Questions:**

Q1: Could the authors elaborate on how their theoretical results might change under heterogeneous or dependent reviewer noise? For example, if reviewers vary in expertise or exhibit correlated biases across papers, would the constant-load result for joint protocols or the Θ(log n / ε) scaling for isolated protocols still hold?

Q2: Could the authors clarify how their framework would extend to settings with strategic or malicious reviewers, such as collusion, biased scoring, or missing reviews?

Q3: Could the author discuss how their main results can be extended to multi-level quality models.

---

### Official Review · Reviewer_CHLn · 2025-11-02

**Soundness:** 3
**Presentation:** 2
**Contribution:** 2
**Rating:** 4
**Confidence:** 3

**Summary:**

This paper presents a formal study of conference review protocols viewed through an information-theoretic and coding-theoretic lens. The authors introduce a general framework for assigning “review tasks” to reviewers and analyzing the accuracy and efficiency of different protocols. They derive asymptotic bounds for isolated, joint, parallel, and adaptive protocols, concluding that isolated protocols have an optimal load ratio of $Θ(\ln (n/ε))$, while joint protocols can achieve constant load independent of both $n$ and $ε$. The work is mathematically nontrivial and introduces valuable analysis techniques (channel coding and Blackwell dominance) to this problem space, with potentially interesting implications for meta-review design and hierarchical evaluation schemes.

The paper demonstrates technical ability and explores a stimulating direction, but the modeling assumptions severely undermine the claimed implications for real peer review systems. The theorems are interesting as abstract results in information theory but do not convincingly apply to practical reviewer assignment or workload design. Significant clarification, redefinition of core variables, and discussion of cost realism are required before the results can be meaningfully interpreted.

**Strengths:**

Ambitious attempt to formalize the efficiency of peer review as an information-theoretic problem.
The technical presentation is sophisticated, and the authors have clearly engaged with prior theoretical work (e.g., Blackwell ordering and adaptive testing).


The results, if correct, offer a formal contrast between isolated and joint protocols that could stimulate further investigation into adaptive or multi-phase peer review.


The framing of peer review as an information aggregation problem is intellectually appealing and connects to recent work in statistical decision theory and crowdsourced evaluation.

**Weaknesses:**

1. Conceptual Modeling Issues
 The central weakness of the paper lies in the abstraction of “review tasks.” Reviewers are modeled as costless information channels that can be assigned arbitrarily complex evaluations over arbitrary subsets of papers. In practice, reviewers incur cost proportional to the number and complexity of papers they read. The model effectively allows a “joint” task to involve reading many papers but charges only a single unit of review cost. This assumption drives the main result that joint protocols can achieve constant load ratios—it seems to be a mathematical artifact of the cost model, not a property of real reviewing systems.
 If reviewer cost were proportional to the number of papers read (as it must be in any realistic setting), the supposed advantage of joint protocols would largely vanish.
2. Ambiguous and Inconsistent Definitions
 Several key quantities are ambiguously defined:
$ε$ appears to represent the probability of making any mistake, not the expected error per paper. This distinction (analogous to FWER vs. FDR) is never made explicit and dramatically affects the asymptotic interpretation.


The “noise parameters” $q_i$​ are described as being drawn from a distribution $D_q$​ over agents (174), but the formalism indexes them by task, implying each task draws an independent noise level. This makes the model equivalent to an infinite pool of independent labelers and removes inter-task correlation—essentially turning the conference problem into a pure crowdsourcing model.
 If this interpretation is intended, it should be made explicit; otherwise, the assumptions and notation should be revised to define noise per reviewer rather than per task.


3. Mathematical and Presentation Ambiguities
The asymptotic notation $Θ(\ln n/ε)$ is written ambiguously and later used interchangeably with $Θ(\ln (n/ε))$. This needs correction.


Theorem 4.5’s claim (specifically the parallel upper-bound; the lower bound seems fine) that joint protocols can achieve load independent of $ε$ seems inconsistent with the $n=1$ special case, where all protocols are tautologically isolated, and estimation reduces to testing the polarity of a Bernoulli, which obviously depends on $ε$. Either the theorem omits conditions on $n$, or the upper bound derivation hides a residual logarithmic dependence.


Figure 1’s axes, colors, and line types are not clearly labeled, and it is unclear what $x,y,z$ represent. The caption for Figure 1D likely contains a typo (“joint and adaptive”).


4. Questionable Interpretations and Motivation
 The “successive review tasks” example illustrates this modeling problem vividly. Each reviewer supposedly checks whether the previous report was correct—a task that, under the model, provides exactly the same information as directly reviewing the paper, while in practice it requires more effort (reading the paper and the prior review). This suggests that isolated adaptive tasks cannot outperform direct evaluation, which is true but trivial. If this is meant to be a theoretical generalization of Blackwell’s theorem for isolated tasks, that connection should be stated plainly.
5. Writing and Style
 The paper is readable but needs proofreading:
“rules for designing review tasks to reviewer” → assigning review tasks to reviewers


“the review load ratio, the number of review tasks per paper” → should use “i.e.,” parentheses, or a colon.


“final estimation’s error probability” → final estimate’s.


Misuse of \citet throughout; many should be parenthetical.

165 Avoid referencing “Section 3” from within Section 3.


169 Typographic style description is a bit nonstandard. “Normal”  may be more clearly described as “Roman” or “upright,” and outcomes (or just variables) should probably be described as “bold(face)” and “bold-italic.”





These issues, combined with mathematical ambiguities, make the work difficult to follow. As a result, the formal results appear more surprising than they actually are—once unpacked, the conclusions mostly restate intuitive facts about the relative information limits of isolated versus non-isolated systems.

**Questions:**

Have you considered allowing for a fixed expected error rate rather than a global error bound $ε$? Would the asymptotic behavior of isolated versus joint protocols remain distinct?


How would your results change if reviewer cost scaled linearly with the number of papers read in a joint task?


Are $q_i$​ known constants or latent random variables? Or are they observed? How are they estimated or incorporated into the design of optimal protocols? This point is very unclear, but they seem to be known, and if so, this is another conceptual limitation, as actually knowing these probabilities seems difficult.


225-231: What’s the point of successive review tasks as defined here? Aren’t the reviewers doing more work (now they have to read the original paper and all the reviews!), only to give the exact same information (by the theoretical model, they are equivalent)? Is the point that any isolated protocol can never ask the reviewers any task that is better than “is the original paper good”? As far as I can tell, the Blackwell argument essentially formalizes this idea over the space of isolated review problems, but in this more narrow domain, it's quite obvious.




*I suspect the answers to the last 2 questions are trivial, but they highlight potential fragility in the distinctions the paper draws between settings, which seems to be a serious limitation.*

---

### Note · Authors · 2025-11-28

**Comment:**

We sincerely appreciate the time and effort the reviewers have dedicated to evaluating our submission, as well as the valuable insights and constructive feedback they provided. After careful consideration, we have decided to withdraw this paper. We are grateful for the opportunity to have participated in the review process and deeply respect the work of the reviewers and the organizing committee.

**Withdrawal Confirmation:**

I have read and agree with the venue's withdrawal policy on behalf of myself and my co-authors.